# Experimental Study on Bond Performance of NC-UHPC Interfaces with Different Roughness and Substrate Strength

**DOI:** 10.3390/ma16072708

**Published:** 2023-03-28

**Authors:** Boshan Zhang, Jiangjiang Yu, Weizhen Chen, Hang Liu, Heng Li, Hong Guo

**Affiliations:** 1Department of Bridge Engineering, Tongji University, Shanghai 200092, China; 2Department of Building and Real Estate, Hong Kong Polytechnic University, Hong Kong 100872, China; 3Shandong Hi-Speed Company Limited, Jinan 250014, China

**Keywords:** interfacial bonding performance, failure evolution, normal concrete, ultra-high performance concrete, interfacial roughness

## Abstract

This paper investigates the bond failure performance between precast normal concrete (NC) and cast-in situ ultra-high performance concrete (UHPC), emphasizing the influence of interfacial roughness. The interfacial bonding behavior under tension and under shear was investigated based on 72 groups of pull-off tests and 36 groups of bi-shear tests, considering six different interface treatment methods and two different NC strength levels. The results demonstrate that certain interfacial roughness is essential to gain a reliable bond connection between NC and UHPC. Its enhancement on the tensile bond performance could be described by the positive linear relationship between the mean roughness and the pull-off strength. However, further research is required to determine the characterization method of its influence on the shear bond performance. The higher strength of the base concrete is beneficial to the bond performance. Though this effect is evident in the pull-off tests under tension, the influence on the shear interfacial performance could be offset by that of the interface morphology in the case with high interfacial roughness.

## 1. Introduction

Concrete is a widely used cement-based material in various civil engineering structures. With the increase in service time, the properties of concrete gradually degenerate, resulting in the declined bearing capacity of the concrete structures or elements. This would seriously affect the safety and durability of existing concrete structures. Considering the huge amount of existing bridges that need to be maintained or strengthened [1], it is critical to developing an efficient, durable, and reliable reinforcement technology [2].

A typical strengthening method is the placement of concrete overlays, where interface bond is a decisive factor, likewise bond properties are crucial for the functioning of precast composite elements [3]. Ultra-high performance concrete (UHPC) has the characteristics of high strength, good toughness, high compactness, and super durability [4,5,6], with broad application prospects in the repair and reinforcement of existing concrete structures or structural members [7,8,9,10]. The statistics in [11] show that there are about 50% of the repaired structural projects have cracks on the bonding surface during the service period. The reliable interfacial bonding performance between UHPC and NC (normal concrete) has become a key factor in forming the desired overall working state, which is improving the performance of a damaged concrete structure to the greatest extent. Regarding the complexity of the interfacial bonding failure between NC and UHPC, it is crucial to study and reveal the interfacial failure behavior between NC and UHPC.

The interfacial behavior under tension is an important index to evaluate the interface bonding performance between the old concrete and the newly cast concrete material [12,13], which is usually tested by the pull-off test. The stress under pure tension is transmitted between the substrate part and the overlay concrete through the physical and mechanical interlocking of the interface and the bonding action of the binder. The loss of the stress transfer mechanism due to the loss of bonding force is a manifestation of premature failure [14]. Fazli [15] conducted 162 groups of pull-off tests and analyzed the bond performance of concrete strengthened with carbon fiber reinforced plastics (CFRP) under the condition of wet/dry cyclic exposure and full-immersion exposure of 3.5% salt-water at a temperature of 60 °C. The results show that after 12 months of environmental exposure, the pull-out strength increased slightly. In the work of Randl [16], the bearing capacity of joints between high-strength concrete (HSC) and normal-strength old concrete (NSC) was investigated, and the influence of sample shape and roughness on joint strength was discussed. Sadowski [17] used principal component analysis (PCA) combined with a self-organizing characteristic diagram to determine the pull-off adhesion between concrete layers, and the genetic algorithm (GA) was adopted to optimize the weight coefficient. Hola [18] conducted a non-destructive evaluation of the pull-off adhesion of concrete layers, and the correlation between the pull-off adhesion strength and the roughness parameters was then established. Tu [19] studied the relationship between macro-scale bond strength and micro-scale ITZ microhardness by using the interfacial bond splitting tensile test and interfacial transition zone (ITZ) microhardness test, and it was found that the splitting tensile strength of interfacial bond has an obvious correlation with the ITZ microhardness. Feng [20] conducted a series of tests to reveal the improvement effect of substrate surface roughness, fiber type, and water/binder ratio on the NC-UHPC bonding behavior. The results show that the improvement of the bonding strength is mainly attributed to the tight ITZ performance of NC-UHPC and the dowel effect of steel fibers on the substrate surface recess. Ghavidel [21] took the self-compacting steel fiber-reinforced concrete as the research object. The optimal value of cement content, the maximum aggregate size, and the steel fiber content affects the pull-off performance was determined.

Besides the tensile load perpendicular to the interface, the NC-UHPC interface in a structural member may also bear the shear action parallel to the interface. There are generally two types of experimental methods to reveal the shear properties of the NC-UHPC interface. The interfacial shear properties under pure shear could be determined by the direct shear test, the push-out test, the bi-surface shear test, and the Z-shaped single-sided shear test. The combined mechanical properties under compression and shear could be evaluated by the slant shear test. Tayeh et al. [22,23,24] carried out experimental research work on the bonding performance of the joint surface between the existing concrete and the cast-in situ UHPC through the slant shear test, the splitting tensile test, and the three-point bending test. However, as the shear failure occurs when the interface overcomes the effect of large friction, the shear strength obtained by this evaluation method is much higher than that of the actual interface. Aaleti and Sritharan [25] evaluated and analyzed the quality of bridge decks considering the shear strength, the interface roughness, and the curing conditions of the substrate concrete. EI-khier [26] carried out small-scale and full-scale bi-surface shear tests to evaluate the interface shear resistance of connectors with the research background of the precast concrete deck system in accelerated bridge construction (ABC). Feng [27] evaluated the bonding performance by the slant shear test with different inclination angles and the splitting tensile strength test. Valipour [28] conducted the general and modified pull-off test and the splitting tensile test to evaluate the bond strength of NC-UHPC slabs, and the bond strength ranged from 2.3 to 3.5 MPa. In Guan’s research [29], push-off tests were carried out on 10 groups of UHPC-NC joint specimens with considerations of the castellated key size and the use of dowel rebar. The grooved surface can be successfully applied to UHPC substrate construction joints because of the significantly improved shear capacity of the interface [30], and proper geometry can produce interlocking effects similar to the aggregates in concrete [31]. Most of the studies on the bonding properties focus on the single shear performance affected by multiple factors without considering the correlation mechanism between the tensile characteristics and the shear performance of the interface.

The roughness of the interface has a positive and significant influence on the bond performance between the UHPC overlay and NC substrate, as the mechanical interlocking of uneven surfaces usually improves the bonding performance [32]. The mechanism of interface linkage can be explained from two aspects: the macro scale and mesoscale [33]. The macro-scale roughness is provided by the texture of the substrate surface, while the roughness mechanism on the mesoscale can be explained as that the newly poured concrete material seeps into the voids or pores of the original concrete surface, forming the mechanical interlocking effect after the concrete setting and hardening [34]. The description of surface roughness depends on the development of various instrument technologies [35]. Current surface morphology detection technologies include the focused ion beam [36], X-ray computed tomography [37], scanning electron microscopy [38,39,40], and confocal microscopy [38,41,42]. Valikhani [43] evaluated the roughness of the substrate surface by using two non-contact test methods: terrestrial laser scanning and digital image processing and correlated the roughness with the bond strength between NC and UHPC. In Randl’s research [44], various shear joint design methods and conceptual models were investigated in detail. It was found that the use of a carefully prepared substrate concrete surface can reduce the number of reinforcements embedded in the interface while not reducing the strength. Zhang [45] investigated the influence of a series of factors on NC-UHPC interfacial tensile performance. The results show that NC surface roughness can significantly improve the interfacial shear capacity, and the number and area of interfacial reinforcement have an enhanced effect on the interfacial shear behavior. In the background of 3D concrete printing, Van et al. [46] proved that lower printing speed leads to higher surface roughness, which further improves the bonding strength between the layers. Ding et al. [35] quantitatively evaluated the fracture surface roughness by a series of parameters such as roughness value, fractal dimension, height distribution standard deviation and arithmetic mean difference of surface morphology. The bi-surface shear study conducted by S.Santos [47] showed that the improvement of surface roughness creates a significant contribution to the achieved bond strength, the bonding agent is advantageous and leads to a higher bonding strength value. However, most of the current researches on roughness measurement concentrated on one type of interface treatment method (chipping, hand nicking, high-pressure water demolition, etc.) in a single study, and there are few studies on the unified comparison of the surface roughness parameters of substrate concrete under different treatment methods.

This paper aims to investigate the interface bonding performance of the NC-UHPC interface, considering the influence of the interface roughness and the base concrete strength. The unified interfacial roughness parameters were determined and obtained by the 3D scanning method, and the interfacial behavior under pure tension and pure shear was tested by the pull-off test and the bi-surface shear test, respectively. The influences of surface roughness and the NC strength were revealed. By taking the mean roughness as the variable, its influence on the tensile bond strength and the shear bond strength of the NC-UHPC interface was analyzed. This study provides a technical basis for practical applications of NC-UHPC composite components or repairing existing concrete structures by UHPC.

## 2. Experimental Program

### 2.1. Materials

In this study, NC of two strength levels, C40 and C50, was adopted to prepare the hardened substrate, and then fresh UHPC was cast on the substrate. The raw materials of NC include the P.C 42.5 Portland cement, the natural river sand with a particle size range of 0–4.75 mm, the basalt aggregates with a particle size range of 4.75–19.00 mm, and the superplasticizer. The specific mix proportions for C40 and C50 concrete are given in Table 1. The mix proportion of UHPC is also given in Table 1, and the raw materials include PII-52.5 cement, silica fume, quartz powder, two kind of quartz sands, superplasticizer, basalt aggregates, and steel fibers. When mixing, the powder materials were first added into the liquids in the order of cement, silica fume, and quartz powder. Then the aggregate was added from fine to coarse, and last the steel fiber was added.

The specimens were cast in two steps, casting NC and casting UHPC. The concrete substrate was first cast and then cured for 28 days. Afterward, the UHPC overlay was cast on the hardened NC substrate, the NC-UHPC specimens were cured for another 28 days, and then the mechanical tests were carried out. The curing was conducted with a temperature of 20 ± 2 °C and humidity over 95%.

The tested properties of NC and UHPC are given in Table 2. 100 mm and 150 mm cubic specimens were used for the compression test of UHPC and NC, respectively. The dog-bone-shaped specimen with a tensile section of 50×50 mm was applied in the tension test of UHPC.

### 2.2. Interface Treatment

Six technical solutions were adopted to treat the NC substrate surface to gain various roughness, as shown in Figure 1. It is worth mentioning that before casting the UHPC overlay, the dust attached to the substrate surface was cleaned up, and then the surface was moistened by water spray. Among them, the treatment named slight roughing (SR, Figure 1a), heavy roughing (HR, Figure 1b), and prick roughing (PR, Figure 1c) were carried out after the initial setting of NC. The other three kinds of chiseled interfaces were treated after 28 days of curing and are named C2 (Figure 1d), C3 (Figure 1e), and C6 (Figure 1f), respectively. The SR solution adopted a serrated roughening blade with a tooth pitch of 5 mm and tooth depth of 3 mm. The HR solution adopted a ribbed bar with a diameter of 10 mm; the notch spacing is 35 mm, and the notch depth is 3 mm. With the PR solution, the hole spacing is approximately 30 mm, and the hole diameter is about 8 mm. The chiseling spacing of C2 is 2 cm, and the average chiseling depth is 3 mm. The chiseling spacing of C4 is 4 cm, and the average chiseling depth is 4 mm. The chiseling spacing of C6 is 6 cm, with an average chiseling depth of 5 mm.

### 2.3. Test Setup

To determine the roughness characteristics, interfaces treated by different solutions were scanned by a Zeiss Smart Zoom 5 microscope. Then the interfacial performance under tension and shear was investigated by 72 groups of pull-off tests and 36 groups of bi-shear tests.

#### 2.3.1. Digital Scanning of Surface

To accurately obtain the roughness parameters of the treated substrate surface, the Zeiss Smart Zoom 5 automated super depth of field digital microscope was used, as shown in Figure 2a. It could measure and compute the roughness parameters automatically. Figure 2b shows the specimen treated with C2 solution, and Figure 2c,d shows the results. Then, some parameters, such as the surface arithmetical mean roughness, *R_a_*, and the surface root mean square roughness, *R_q_*, were obtained. Among them, *R_q_* is not significantly influenced by scratches, contamination, and measurement noise. The specific steps of measurement and calculation are as follows:

(1) Preparing the substrate concrete samples with different roughness (testing area of 100 mm × 100 mm) and cleaning the surface dust;

(2) Placing the sample on the platform of the digital microscope and selecting 50 × Lens, automatic scanning after focusing;

(3) Automatic scanning, recording, and reconstructing substrate surface with 5 μm slice height;

(4) According to Equation (1) (proposed by Santos et al. [48]), the arithmetic mean roughness *R_a_* can be calculated as follows:(1)$$R_{a}=\frac{1}{A}\iint_{A}\left|z(x,y)\right|dxdy$$
where $\left|z(x,y)\right|$ is the absolute values of peaks and valleys measured from the optimized plane, and *A* is the measuring area.

(5) The root mean square roughness *R_q_* can be calculated as follows:(2)$$R_{q}=\sqrt{\frac{1}{A}\iint_{A}z^{2}(x,y)dxdy}$$

#### 2.3.2. Test Method for Interfacial Mechanical Properties

The pull-off test on drilled core samples was used to characterize the bonding properties between NC and UHPC under direct tension. As Figure 3a shows, a core sample was drilled on a composite slab that consists of an 80 mm thick NC substrate and a 40 mm thick UHPC overlay. The diameter and depth of the sample are 55 mm and 60 mm, respectively. Then, a steel disk was stuck to the drilled samples of the two-component epoxy resin adhesive. For each interface type, six samples were tested. When loading, the tension force was applied and measured by the HC-6000C tester, as seen in Figure 3b.

The bi-shear test was used to reveal the interfacial failure behavior under shear. The 150 mm × 150 mm × 150 mm cubic specimen is composed of two parts. The dimension of the UHPC part is 150 mm × 150 mm × 50 mm, and that of the NC part is 150 mm × 150 mm × 100 mm, as shown in Figure 4a. The specimen was loaded, as Figure 4b shows. Two linear variable differential transformers (LVDTs) were arranged to measure the interface slips, and two LVDTs were arranged to measure the global displacement. The displacement-controlled loading mode with a speed of 0.2 mm/min was adopted. The data acquisition frequency was 5 Hz.

## 3. Experimental Results

### 3.1. Results of Surface Scanning

The obtained arithmetical mean roughness, *R_a_*, and the surface root mean square roughness, *R_q_*, of various interfaces treated by different methods are given in Figure 5. The results show that the surface roughness with SR treatment is the smallest, with *R_a_* = 0.29 and *R_q_* = 0.41. The surface roughness with HR treatment is the largest, with *R_a_* = 0.96 and *R_q_* = 1.31, which are 3.31 times and 3.19 times, corresponding to SR, respectively. It can also be seen from Figure 1 that the fluctuation of the SR surface is the smallest, while that of the HR surface is the roughest.

When the interface between UHPC and NC is subjected to shear stress, the interface of HR treatment can provide a greater interlocking effect. The *R_a_*/*R_q_* values of C4, C2, PR, and C6 interfaces are 0.36/0.47, 0.48/0.66, 0.48/0.91, and 0.61/0.95, respectively. Among the three chiseling surfaces, the C6 surface gains larger roughness though there are few chiseling points as the larger depth and area of the chiseled points. The calculated *R_a_* of C2 and PR are close; however, *R_q_* of the PR treatment specimen is significantly larger than C2, which indicates that the fluctuation dispersion degree of the substrate concrete surface morphology with PR treatment is greater than C2.

### 3.2. Experimental Results of the Pull-Off Test

Based on the pull-off tests, the bond behavior between NC and UHPC under tension was tested and analyzed.

#### 3.2.1. Failure Mode

For the pull-off test sample under pure tension, there are four possible failure modes, as shown in Figure 6: (a) failure at the bond interface, (b) failure in the substrate, (c) failure in the overlay, and (d) failure at the adhesive layer between the overlay and the pulling disk. In this research, failure mode (c) would not happen due to the high strength of the UHPC overlay. With mode (d), the result is invalid, and the sample should be tested again.

Two failure modes, (a) at the interface and (b) in the NC substrate, have been recorded in the conducted pull-off tests, and the numbers of each pull-off failure mode for different types of samples are listed in Table 3. For samples with C40 substrate, interfacial failure mode (a) occurs in 29 samples, and NC failure mode (b) occurs in four samples. In comparison, the numbers change to 11 and 24 for samples with C50 substrate.

Figure 7 shows the failed pull-off samples and their fracture surface, where the samples with C40 substrate are shown in Figure 7a1–f1, and the samples with C50 substrate are shown in Figure 7a2–f2. It can be seen that the coarseness of the fracture surface varies significantly with the substrate strength level. For most samples with C40 substrate, only a small amount of hardened concrete mortar is attached to the UHPC core, and the failure surface is relatively smooth where the traces of interface treatment are visible. It is consistent with the features of the interfacial failure mode (a). While, for most samples with C50 substrate, the broken concrete is attached to the UHPC, and exposed aggregates can be seen at the coarse failure surface, which indicates the failure in NC.

#### 3.2.2. Pull-Off Strength

The nominal pull-off strength, *T_s_*, could be calculated by Equation (3) as
(3)$$T_{s}=\frac{P_{ut}}{A_{t}}$$
where *P_ut_* is the ultimate tensile force at failure, and *A_t_* is the section area of the drilled-core sample. The pull-off strength reflects the interfacial bond strength for failure mode (a) and the tensile strength of NC for failure mode (b). In mode (b), the unmeasured interfacial strength is higher than the pull-off strength.

Table 3 lists the average pull-off strength of various types of samples and the corresponding standard deviations and coefficients of variance. The results are sorted by the substrate type and the interface roughness. It is apparent that the pull-off strength is affected by both the substrate strength level and the interface roughness. For samples with the same interface, the average pull-off strength improves when C40 concrete is replaced by C50 concrete as the substrate. For samples with the same substrate, the pull-off strength rises with the interface roughness. The pull-off strength reaches the maximum value when the interface is treated by the HR method with the highest roughness, and the value is 2.25 MPa with the C40 substrate and 3.43 MPa with the C50 substrate. The minimum value appears in samples treated by the SR method with the lowest roughness, and the value is 0.96 MPa with C40 substrate and 2.09 MPa with C50 substrate. It is worth mentioning that no failure mode (b) was observed for the C40 substrate specimen corresponding to the HR interface treatment. The reason is that for the pull-off test, the plane size of the prefabricated test slab is relatively large compared with the sampling area of the drill-core specimens, and the chiseling spacing and depth corresponding to the HR treatment method are also relatively large compared with other treatment methods (it can be seen from Figure 7). Therefore, under the internal tensile load of the small circular bearing area, the interface stress distribution is relatively uniform, and the debonding failure is more likely to occur on the interface. The max and the min pull-off strength increases, respectively, by 116% and 52% when the substrate strength level increases. For the other four types of samples, the growing range is 149% for samples treated by the C4 method, 81% for samples treated by the C2 method, 71% for samples treated by the PR method, and 52% for samples treated by the C6 method. In addition, the samples with C40 substrate always gain higher variation coefficients.

### 3.3. Experimental Results of The Bi-Shear Test

Based on the bi-shear tests, the bond behavior between NC and UHPC under shear was tested and analyzed.

#### 3.3.1. Failure Mode

In the bi-shear test, the NC-UHPC interface and its symmetrical plane in NC are regarded as the two concerned shear planes as they bear the maximum vertical shear equal to half of the applied loading force, as demonstrated in Figure 8. Depending on the shear strength of NC, UHPC, and the NC-UHPC interface and the relationship between them, there are supposed to be three failure modes: (a) failure at the NC-UHPC interface, (b) failure in NC and at the interface, and (c) failure in NC. When the interface shear strength is lower than those of NC and UHPC, failure mode (a) occurs. The damage could only initiate and develop along the NC-UHPC interface and eventually leads to interfacial failure. Failure mode (c) tends to occur when the interface shear strength is higher than that of the concrete, in which cracking generally generates in NC near the two concerned shear planes. Failure mode (b) is between the former two modes, where the interfacial strength and the NC strength under shear are similar.

In the conducted bi-shear tests, two types of failure modes, mode (a) and mode (b), have been observed, as shown in Figure 9a,b. Mode (a) accounts for the cases where the NC-UHPC interface totally deboned suddenly. In mode (b), micro-cracks appeared both in the NC part and at the NC-UHPC interface and then developed and extended in the meantime. Due to the heterogeneity of concrete and the randomness of the internal defects, the cracking trajectory also varies, leading to one oblique fracture plane or two vertical shear fracture planes. The numbers of each bi-shear failure mode for different types of specimens are listed in Table 4. For samples with C40 substrate, failure mode (a) occurs in 6 specimens, and failure mode (b) occurs in 12 specimens. In comparison, the numbers change to 1 and 17 for specimens with C50 substrate.

For NC-UHPC interfaces with a specific treatment, the damaged interfaces show a similar appearance, though with different substrate strength levels. Figure 10 shows the typical interfaces after failure with various treatment methods. It can be seen that the coarseness of the fracture surface is closely related to the treatment method, and a positive correlation could be observed between the coarseness of the fracture surface and the interface roughness. As Figure 10b,c show, the interfaces treated PR and HR gain relatively high roughness, and there is an NC layer attached on the UHPC part when failed. Additionally, there is less attached NC for interfaces with small roughness. For the three types of chiseled roughened interfaces, the damaged interfaces are similar, as seen in Figure 10d. The roughened pit point can provide a certain level of interlocking effect, beneficial to the interface strength under shear. When the interfacial failure occurs, the UHPC bulges located in the pit points break, and there is barely NC attached to the UHPC part.

#### 3.3.2. Bi-Shear Strength

The nominal bi-shear strength, *B_s_*, could be calculated by Equation (4) as
(4)$$B_{s}=\frac{P_{ub}}{2A_{\tau }}$$
where *P_ub_* is the ultimate force applied on the bi-shear specimen, and *A_τ_* is the area of the fracture section. The bi-shear strength reflects the interfacial bond strength for failure mode (a) and mode (b).

Table 4 lists the average bi-shear strength of each type of specimen in the order of the roughness value, and the corresponding standard deviations and the coefficients of variance are also provided. The results are sorted by the substrate type and the interface roughness. It could be seen that the bi-shear strength varies with the substrate strength level and the interface roughness. For specimens with the same interface treatment method, the average bi-shear strength increases after replacing C40 concrete substrate with C50 concrete substrate, and the substrate concrete strength has a positive effect on the bi-surface shear strength. For specimens with C40 substrate, the maximum value is 5.02 MPa with C6 treatment, and the minimum value is 2.37 MPa with C2 treatment. For specimens with C50 substrate, the maximum value is 4.37 MPa with C2 treatment, and the minimum value is 2.41 MPa with PR treatment.

## 4. Discussions

### 4.1. Influence of Substrate Strength and Interfacial Roughness

#### 4.1.1. Influence on Pull-Off Strength

In Figure 11, the proportion values of certain kinds of failure modes in different tested sample groups are calculated, and the results are arranged in sequence of the interfacial roughness. Generally, the proportion value of failure at the bond line decreases as the substrate strength level grows from C40 to C50, and the average proportion value reduces from 88% to 31%. It indicates that the tensile bonding performance between NC-UHPC improves since the improvement of substrate strength level, and moreover, the increased range of the interfacial strength exceeds that of the substrate for most cases. Though the reduction degree varies with the interface treatment, ranging from 33.3% to 100.0%, no regular relationship has been observed with the interface roughness.

In Figure 12, the individual pull-off strength of all samples and the average values with C40 and C50 substrates are plotted, respectively. The results are sorted in order of interface roughness. For both, the average pull-off strength rises as the interface roughness grows from 0.29 to 0.96. Then, the Pearson correlation coefficient between the interface roughness and the pull-off strength is obtained by Equation (5) as
(5)$$r=\frac{\sum_{i=1}^{n}\left(X_{i}-\bar{X}\right)\left(Y_{i}-\bar{Y}\right)}{\sqrt{\sum_{i=1}^{n}{\left(X_{i}-\bar{X}\right)}^{2}\sqrt{\sum_{i=1}^{n}{\left(Y_{i}-\bar{Y}\right)}^{2}}}}$$
where *X_i_* and *Y_i_* are the individual values of the variables, $\bar{X}$ and $\bar{Y}$ are the mean values of the variables, and *n* is the number of samples of each variable. The correlation coefficient between the interface roughness and the pull-off strength is 0.93 for samples with C40 substrate and 0.95 for samples of C50 substrate. There is a strong positive correlation between the interface roughness and the pull-off strength. Thus, the linear fitting has been conducted to reveal the relationship between them, where the interface roughness is regarded as the independent variable,
(6)$$T_{s}=aR_{a}+b$$
where *a* is the slope of the fitting curve, and *b* is the intercept.

The linear fitting results of pull-off strength and roughness are shown in Figure 12 and Table 5. The goodness-of-fit is high, as both of the coefficients of determination, *R*^2^, are over 0.85. Positive slopes were gained for both fittings, and the slope of the C50 case is slightly lower than that of the C40 case. Within the investigated range of interface roughness, the interface roughness level has an obvious enhancement effect on both the pull-off strength and the consistency of the results, and this effect is slightly obvious when the base strength is low.

The intercept difference of the two fitting curves reflects the significance of the substrate strength to the interfacial bond performance between NC and UHPC. An investigation [20] on the bond properties between UHPC and NC substrate proposed that microcracks are prone to generate around the overlay transition zone (OTZ) under tensile force, leading to the debonding failure between the cast-in situ UHPC and the base concrete. The base concrete with a higher strength level is conducive to better microstructure and macro-performance of OTZ, thus improving the bonding performance of the NC-UHPC interface. Combined with the tendency of the failure mode in samples with different substrates reflected by Figure 11, it is reasonable to predict that the enhancement effect of the substrate strength on the interface strength under tension would be greater than the results reflected by the above values.

#### 4.1.2. Influence on Bi-Shear Strength

In Figure 13, the bi-shear strength of each specimen and the corresponding average values of different types of specimens are plotted. The results are sorted in order of the mean interfacial roughness. For specimens with different substrate strengths, the variation trends of the bi-shear strength with the mean interfacial roughness are consistent, which shows that the interface morphology has a regular influence on the bi-shear strength.

Different from the monotonic increase in the pull-off strength with the interface roughness, the bi-shear strength decreases when the mean roughness increases to about 0.5, and then it increases with the mean roughness. The correlation coefficient between the bi-shear strength and the mean roughness was calculated by Equation (5), and the result is 0.68 for specimens with C40 substrate and 0.05 for specimens with C50 substrate. The correlation degree is significantly lower than that of the pull-off strength and the mean roughness, especially in the case of the C50 substrate. Linear relation could not reflect the relationship between the variables well, indicating that the effect of interface morphology on bi-shear strength cannot be reasonably characterized by the mean roughness, *R_a_*.

Generally, the interfacial strength between NC and UHPC is the comprehensive embodiment of the adhesion, friction, and interlocking action [49]. The pull-off strength under pure tension is mainly provided by the adhesion between NC and UHPC [50], and the interface morphology mainly affects the bonding area. Thus, the mean roughness, *R_a_*, can feature its influence on the pull-off strength. Moreover, the bi-shear strength is affected by the adhesion action and interlocking action, and the effect of the interface morphology is more complex. It requires further exploration to determine the appropriate characterization method.

Additionally, it should be noted that the bi-shear strength with the C40 base would exceed that with the C50 base when the mean roughness *R_a_* is over 0.5. The blue dash line in Figure 13 shows the strength ratio between them. The ratio is over 1.0 in cases with SR, C4, and C2 treatment while lower than 1.0 in cases with PR, C6, and HR treatment. This strength reduction in counter commonsense in specimens with higher base strength levels is due to the diversity of the manually treated interface morphology. With manual treatment, the interface consisting of each specimen cannot be guaranteed. It also shows that the interface morphology rather than the base strength becomes the main factor affecting the bi-shear strength with the increase in interfacial roughness.

### 4.2. Load–Slip Characteristics under Shear

The load-interface slip curves of the bi-shear specimens with various interface treatments are plotted in Figure 14a–f. There are two typical forms of load-interface slip curves corresponding to two failure modes specified in Figure 9.

The first type of curve corresponds to the failure mode (a) shown in Figure 9a, where the NC-UHPC interface debonded suddenly. The curves of specimens SR-C40-2 and SR-C50-2 are two examples. This type of curve experiences two stages, the linear stage, and the brittle failure stage. Before cracks initiate at the NC-UHPC interface, the slip value grows linearly with the load. The slip value in the linear stage is caused by the different Young modulus of the NC component and the UHPC component. Thus, the value is very small and usually negligible. When the ultimate load is reached, brittle debonding failure occurs at the interface.

The second type of curve accounts for the failure mode (b) shown in Figure 9b, and the examples include the curves of the specimens treated by the C4 method. This form of curve includes three stages: the linear stage, the strengthening stage, and the yielding stage. The characteristics of the linear stage are consistent with that of the first type of curve. Then, due to the initiation and development of cracks at the interface and in NC, the slope of the load–slip curve decreases as the decrease in shear stiffness, and it enters the strengthening stage. At this stage, the sliding value increases obviously while the load continues to increase. When the damage accumulates to a certain extent, the curve enters the yield stage, where the interface slip increases rapidly, with the load remaining around the critical value until the sudden fracture occurs. For engineering applications, it is beneficial to ensure structural safety if the enhancement in the strengthening stage can be improved with a similar strength level to the linear stage.

## 5. Conclusions

In this study, the debonding failure characteristics of NC-UHPC interfaces with different interface treatment methods and substrate concrete strength levels were investigated on the basis of a series of pull-off tests and bi-shear tests. The interface roughness parameters were adopted to analyze the influence of interface morphology introduced by different treatment technical solutions. The following conclusions can be obtained:(1)The NC-UHPC interface could gain good bond performance when the surface of the concrete base is treated by proper roughening methods. The pull-off strength under tension reaches the maximum value when adopting the heavy roughing (HR) method, and the value reads as 2.25 MPa with a C40 base and 3.34 MPa with a C50 base. The maximum bi-shear strength under shear is 5.02 MPa with a C40 base and 4.37 MPa with a C50 base, where two chiseling methods are used.(2)The interfacial roughness formed by treatment on the base surface has an obvious enhancement effect on the pull-off strength of NC-UHPC interfaces under tension as the adhesion area increases. This influence could be characterized by the mean roughness, *R_a_*, which shows a positive linear correlation with the pull-off strength.(3)The interface morphology has a regular influence on the interfacial strength between NC and UHPC under pure shear. It cannot be characterized by the mean roughness, *R_a_*, and further research is required on the characterization method.(4)Since a higher base concrete strength level helps to form better microstructure and macro-performance of the overlay transition zone, it also contributes to the bond performance between NC and UHPC. This effect is evident in the pull-off tests under tension. In the bi-shear tests, the interface morphology becomes the dominant affecting factor as the interface roughness increases, whose influence would offset that of the substrate strength.(5)The bi-shear failure process could be divided into two categories according to the characteristics of the load–slip relationship and the difference lies in the existence of the strengthening phase after the linear stage. Considering engineering applications, it is beneficial to ensure structural safety if the loading enhancement in the hardening stage can be improved.

## Figures and Tables

**Figure 1 materials-16-02708-f001:**
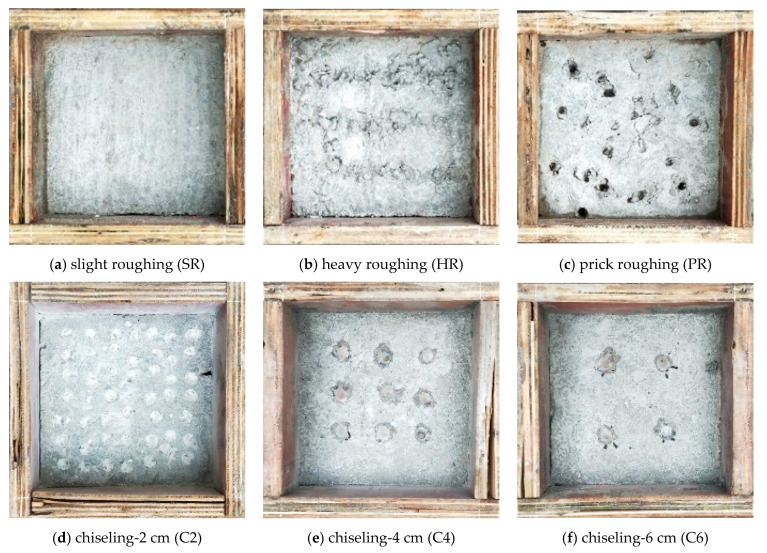
Interface treatment.

**Figure 2 materials-16-02708-f002:**
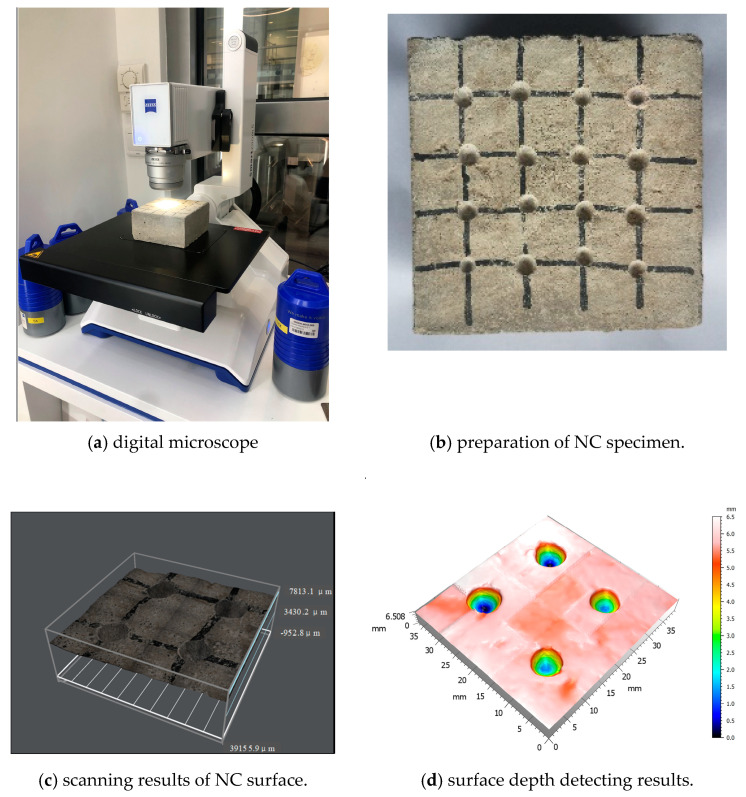
Scanning of the treatment interface.

**Figure 3 materials-16-02708-f003:**
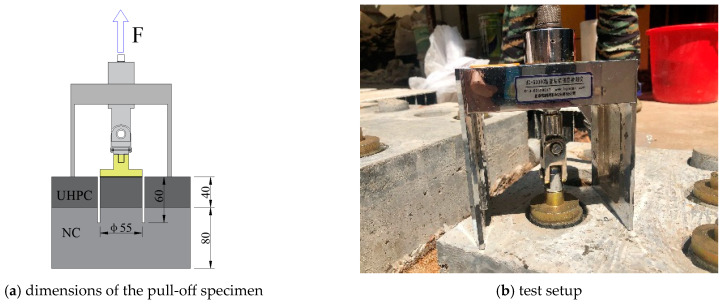
Layout the pull-off test (unit: mm).

**Figure 4 materials-16-02708-f004:**
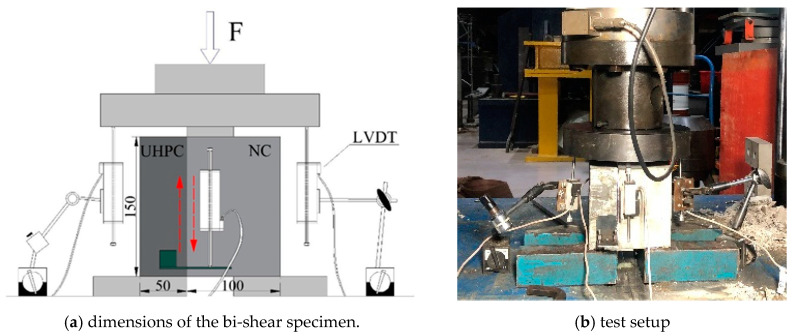
Layout of the bi-shear test (unit: mm).

**Figure 5 materials-16-02708-f005:**
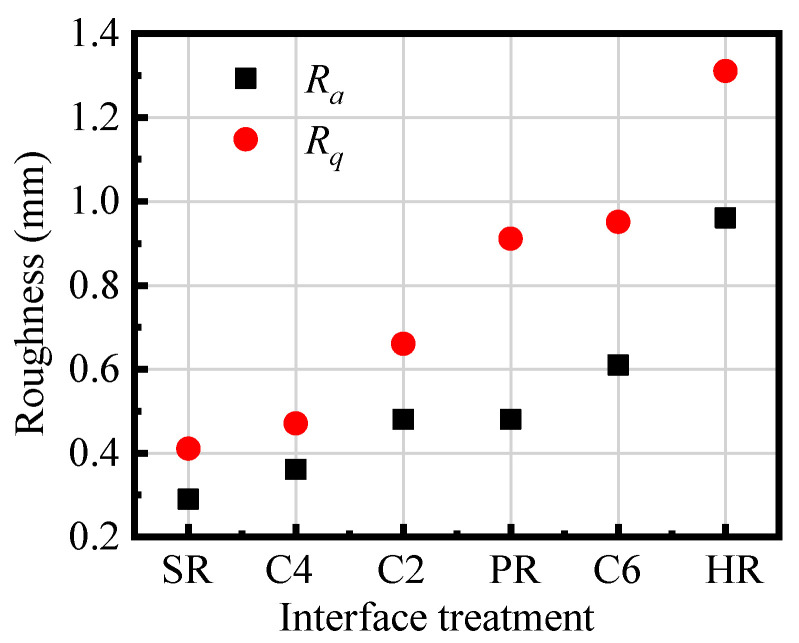
Measured roughness parameters of interfaces treated by different methods. Note: *R_a_* is the arithmetical mean roughness, and *R_q_* is the surface root mean square roughness.

**Figure 6 materials-16-02708-f006:**
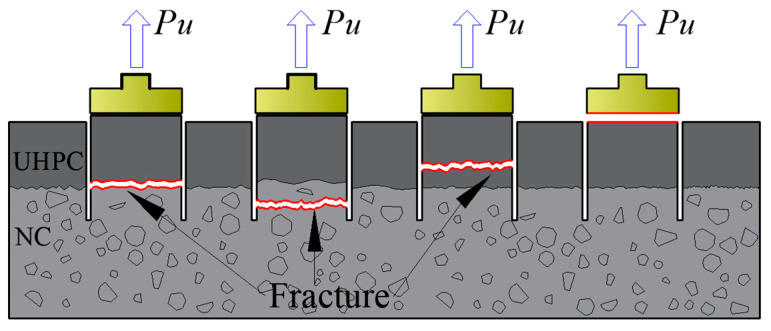
Possible failure modes of the pull-off test.

**Figure 7 materials-16-02708-f007:**
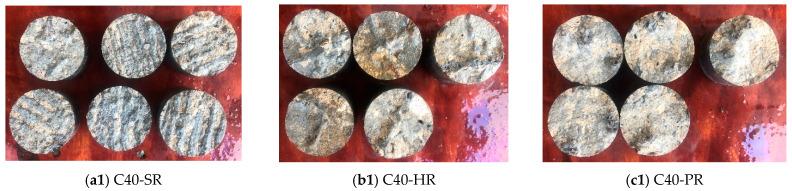
Fracture sections of the pull-off samples.

**Figure 8 materials-16-02708-f008:**
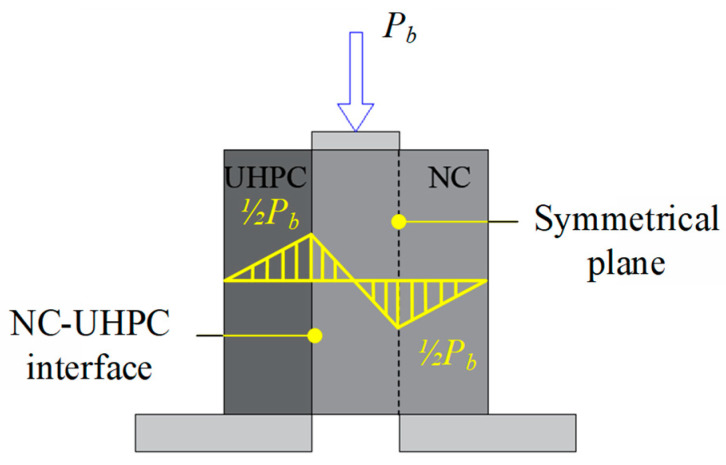
Internal shear force in the bi-shear specimen.

**Figure 9 materials-16-02708-f009:**
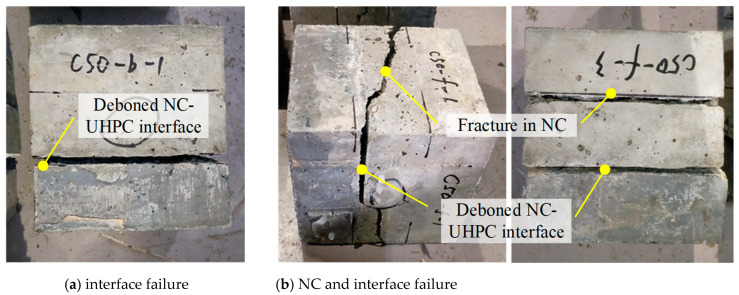
Failure modes of bi-shear tests.

**Figure 10 materials-16-02708-f010:**
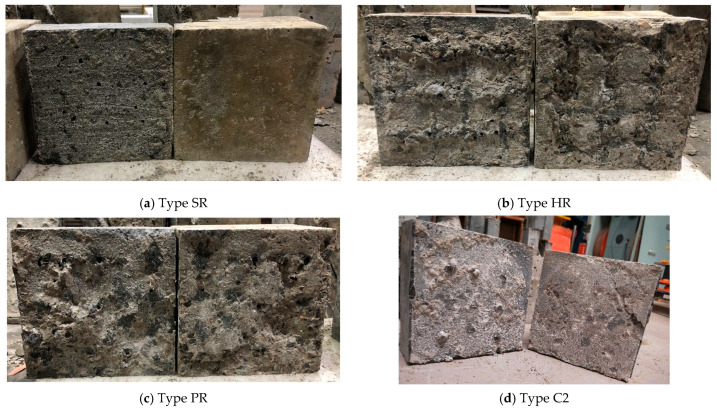
Failed interfaces with different interface processing methods.

**Figure 11 materials-16-02708-f011:**
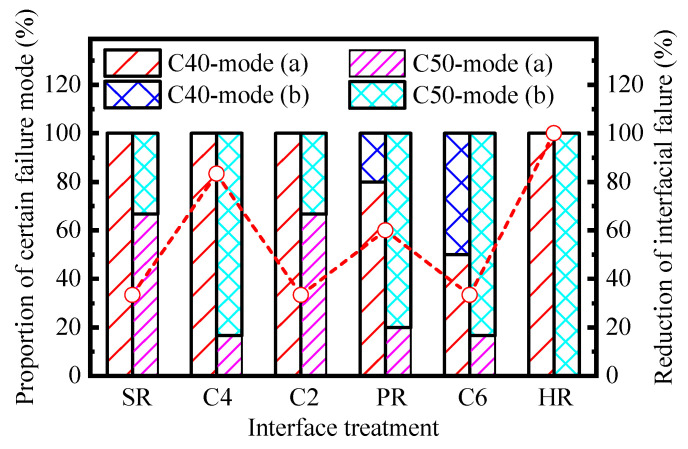
Statistics results of pull-off failure positions.

**Figure 12 materials-16-02708-f012:**
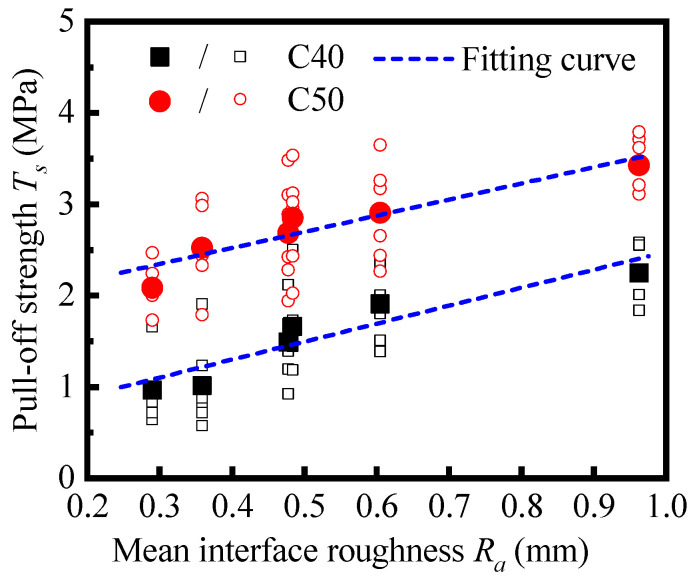
Relationship between pull-off strength and interface roughness *R*_a_.

**Figure 13 materials-16-02708-f013:**
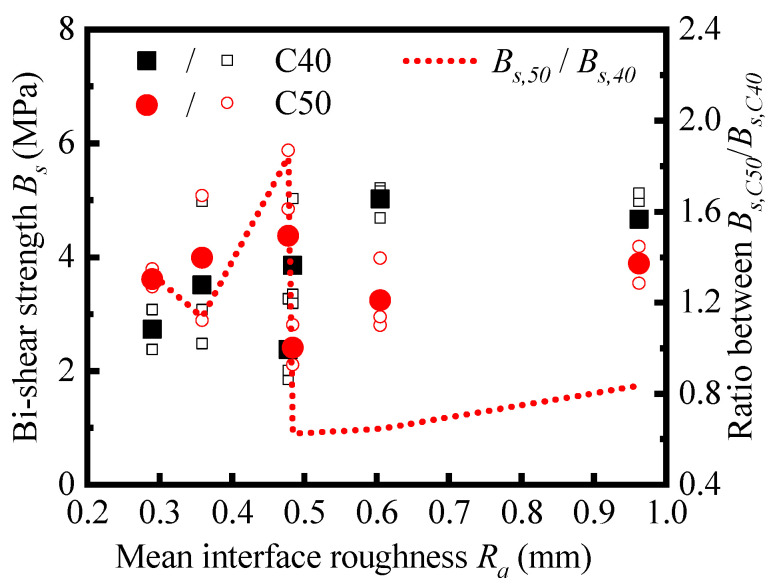
Relationship between bi-shear strength and interface roughness *R_a_*.

**Figure 14 materials-16-02708-f014:**
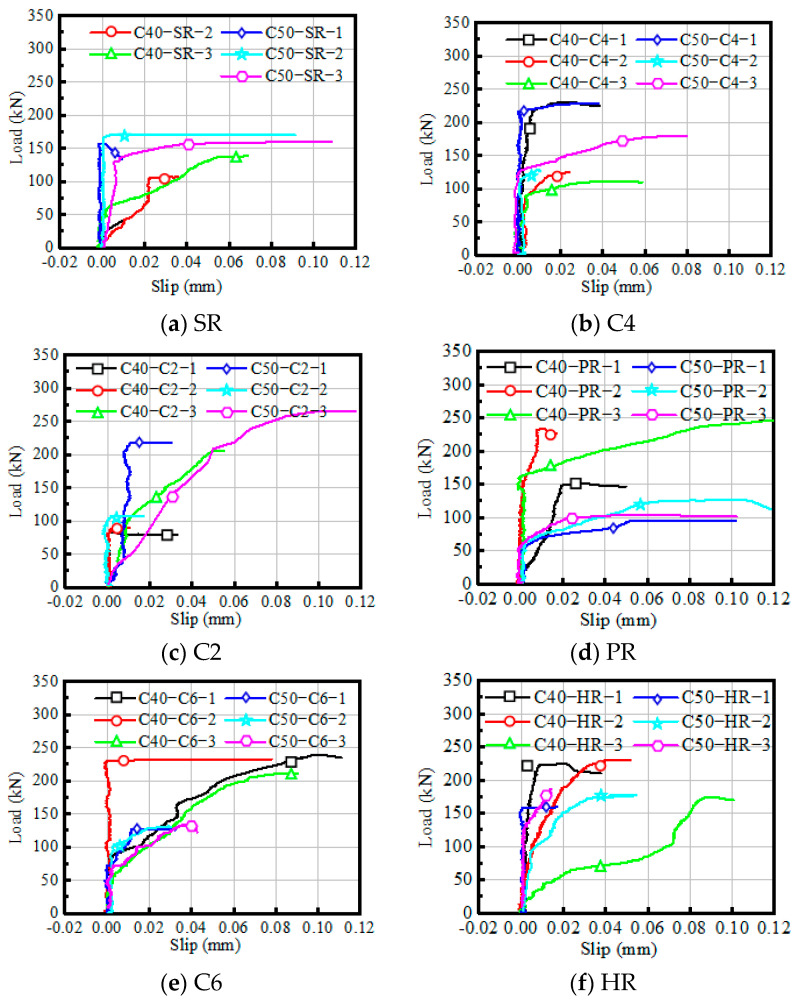
Relation between the load and the interface slip.

**Table 1 materials-16-02708-t001:** UHPC [5] and NC mixtures (kg/m^3^).

Material	Raw Material	Amount (C40/C50)
NC	Cement (42.5R)	350.0/420.0
	Water	175.0/170.0
	Superplasticizer	7.1/8.2
	Sand	750.0/680.0
	Coarse aggregate	1040.0/1060.0
UHPC	Cement (52.5R)	680.0
	Silica fume	81.6
	Quartz fines	272.0
	Quartz sand (0.15–0.20 mm)	169.8
	Quartz sand (0.25–0.60 mm)	433.6
	Superplasticizer	17.0
	Water	159.8
	Basalt aggregate	581.0
	Steel fiber	117.0

**Table 2 materials-16-02708-t002:** Mechanical properties of UHPC and NC.

	NC(C40)	NC(C50)	UHPC
Elastic Modulus (MPa)	33,500	34,500	44,500
Poisson’s ratio	0.20	0.20	0.18
Compressive strength (Mpa)	45.7	56.1	135.5
Tensile strength (Mpa)	2.9	3.2	9.1

**Table 3 materials-16-02708-t003:** Summary of the pull-off test results.

Substrate Type	Interface Treatment	Mean Pull-Off Strength (MPa)	Number of Failed Samples		Standard Deviation (MPa)
			Mode (a)	Mode (b)	
C40	SR	0.96	6	0	0.36
	C4	1.01	5	0	0.45
	C2	1.49	6	0	0.38
	PR	1.66	4	1	0.41
	C6	1.91	3	3	0.39
	HR	2.25	5	0	0.27
C50	SR	2.09	4	2	0.23
	C4	2.52	1	5	0.57
	C2	2.69	4	2	0.52
	PR	2.85	1	4	0.49
	C6	2.91	1	5	0.49
	HR	3.43	0	6	0.29

**Table 4 materials-16-02708-t004:** Summary of the bi-shear test results.

Substrate Type	Interface Treatment	Number of Failed Specimens		Mean Bi-Shear Strength (MPa)	Standard Deviation (MPa)
		Mode (a)	Mode (b)		
C40	SR	2	1	2.73	0.35
	C4	0	3	3.51	1.07
	C2	1	2	2.37	0.08
	PR	1	2	3.85	0.83
	C6	1	2	5.02	0.24
	HR	1	2	4.66	0.56
C50	SR	1	2	3.61	0.13
	C4	0	2	3.99	0.89
	C2	0	3	4.37	1.23
	PR	0	3	2.41	0.29
	C6	0	3	3.24	0.52
	HR	0	3	3.89	0.27

**Table 5 materials-16-02708-t005:** Fitting results for the pull-off results.

Substrate Type	*a*	*b*	Coefficient of Determination *R*^2^
C40	1.9655	0.5056	0.87
C50	1.7619	1.8124	0.89

## Data Availability

Data is contained within the article.

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
