# Peer review of "Experimental Study on Bond Performance of NC-UHPC Interfaces with Different Roughness and Substrate Strength"

_materials, 2023, doi:10.3390/ma16072708_

Round 1

Reviewer 1 Report

The manuscript provides interesting results on NSC-UHPC interface which meet the expectations concerning observed bond strength values. The following comments should be tackled before publication:

Introduction:

Line 34, a sentence should be added to provide the linkage between strengthening and interface bond, i.e. between first and second paragraph. For example, the Authors could say the following: A typical strengthening method is the placement of concrete overlays, where interface bond is a decisive factor, likewise bond properties are crucial for the functioning of precast composite elements [1].

[1] J. Liu et al. (2020), Experimental study on interfacial shear behaviour between ultra-high performance concrete and normal strength concrete in precast

composite members, Construction and Building Materials 261 (2020): 120008

Page 2: Series of push-out shear and pull-off tensile tests have recently been performed in a comprehensive study investigating bond between high strength concrete (HSC) overlay and NSC substrate, this study should also be mentioned in that context:

[2]  Randl, N., Steiner, M., Peyerl, M. (2020), Hochfester Aufbeton zur Tragwerksverstärkung: Teil 1: Kleinkörperversuche, Beton- und Stahlbetonbau, Vol. 115 (2), p. 106-116, https://doi.org/10.1002/best.201900028

Page 3 – linkage between roughness and bond performance: It is correct that the Authors point out the additional effect of mechanical interlocking the rougher the interface. The additional interacting effect of such mechanical interlocking and its quantitative contribution has already been analyzed in the past on the basis of test results in [3] which should be mentioned.

[3]  Randl, N., Zilch, K., Müller, A. (2008), Bemessung nachträglich ergänzter Betonbauteile mit längsschubbeanspruchter Fuge, Beton- und Stahlbetonbau, Vol. 103 (7), p. 482 – 497, https://doi.org/10.1002/best.200800627

Line 148: „our“ – avoid possessive adjectives in a technical paper

Lines 150-157:

-        Mention the concrete’s age at the day of testing

-        Shape and size of compressive specimens?

Interface treatment:

Apart from roughness, please describe whether there was any other additional treatment before placing the UHPC overlay (e.g. cleaning, moistening … ? )

Experimental results: The term “dowel action” is not correctly used here – actually dowel action always means in such context the bending resistance of connectors. The Authors mean probably the interlocking effect? Same applies also in line 327.

Table 3: The Authors correctly point out the effect of substrate strength and interface roughness on pull-off strength. It is on the other hand surprising that failure mode (b) is with C40 substrate not observed with HR-interface, do the Authors have any explanation?

Line 357: “interfacial strength exceeds the that of the substrate” – delete “the“

Line 372: probably the Authors mean pull-off and not pull-out strength?

Lines 379 to 383: Such saying is only valid for the investigated range of roughness. In fact, it can be observed that with further increasing roughness (mean roughness > 2 mm), such effect decreases (as observed e.g. in [2]). Please mention such limitation of the derived linear fitting.

Line 387: “substrate” instead of “substate”

Page 15: “the bi-shear strength decreases sharply …” this saying is based on some single tests which is not sufficient to state it that strongly, and it is also due to the type of the introduced fitting curve. Please re-formulate in order not to over-emphasize the observation.

Lines 413/414: The Bi-shear test does not create friction effects, please correct.

“pin action”: do the Authors mean interlocking effect?

Page 16 and conclusions: The term “strengthening stage” is not common, please use “hardening stage” instead.

Conclusion 3: “Due to the complexity of the law of action, it …” The explanation is neither well based nor required, therefore it is recommended to omit it and just start the sentence “It cannot be characterized by the mean roughness …”

Typos in Figures:

Fig. 9 – probably the Authors mean debonded instead of deboned

Fig. 11 “Probability of centain failure mode” – Authors probably mean “certain”,      “Reduction of interfacial falure” – Authors probably mean “failure”

Reviewer 2 Report

The focus of this paper is the bond failure performance between precast normal concrete (NC) and cast-in-situ ultra-high-performance concrete (UHPC), with an emphasis on the impact of interfacial roughness. Although the article is pertinent to the scope of the journal and presents interesting findings, there are some areas that require improvement for it to be considered for publication in the Materials journal.

Specific comments:

1)        Could you please provide more information on the mixing process used, given that the cement consumption for the UHPC is comparatively low and that coarse aggregate is present in the resulting mixture? (Line 142)

2)        About the Digital scanning of surface, the software utilized for this purpose publicly is available on the internet, and what is the typical input image format used for this process? (Line 182)

3)        To prevent any potential confusion, it is recommended that the authors include the term "crack" in Figure 6.

4)        What was the method utilized by the authors to attach the metal plate to the UHPC? Please, provide this information.

5)        Did the authors measure the relative displacement between the surfaces of UHPC and NC (test 1)? Is this result significant as input for numerical simulations?

6)        The authors should provide the stress-strain diagrams for both tension and compression for NC and UHPC.

7)        What is the reason for the observed variability in the results, and have the authors conducted any outlier tests?

8)        What is the reason for the lower standard deviation observed in the C50 samples compared to the C40 samples?

9)        Was normality checked in the results?

1)        In Figure 13, what does the dashed blue line specifically represent?

2)        I disagree with the term "probability of failure," as obtaining this parameter would require a much greater number of results. In Figure 11, it would be more appropriate to display the frequency of observation of the failure modes.

Round 2

Reviewer 1 Report

The comments have been considered by the Authors and the manuscript deems appropriate for publication.

The Authors are just asked to correct the following minor editorial mistake in Line 293:

“It is worth mentioning that there is no failure mode (b) was observed for the C40 substrate specimen corresponding to the HR interface treatment.”

The phrase “there is” is redundant and should be deleted.

Author Response

Thanks to the comment, and the phrase "there is" has been deleted from the sentence in line 293.

Reviewer 2 Report

Accept

Author Response

Thank you very much for your approval of acceptance.